# The Nephroprotective Effects of the Allogeneic Transplantation with Mesenchymal Stromal Cells Were Potentiated by ω3 Stimulating Up-Regulation of the PPAR-γ

**DOI:** 10.3390/ph16101484

**Published:** 2023-10-18

**Authors:** Andreia Silva de Oliveira, Márcia Bastos Convento, Clara Versolato Razvickas, Bianca Castino, Ala Moana Leme, Rafael da Silva Luiz, Wesley Henrique da Silva, Maria Aparecida da Glória, Tatiana Pinotti Guirão, Eduardo Bondan, Nestor Schor, Fernanda Teixeira Borges

**Affiliations:** 1Translational Medicine Division, Department of Medicine, Federal University of Sao Paulo, São Paulo 04038-901, Brazil; andreiaoliveira9921@gmail.com; 2Nephrology Division, Department of Medicine, Federal University of Sao Paulo, São Paulo 04038-901, Brazil; convento@unifesp.br (M.B.C.); claraversolato@gmail.com (C.V.R.); alamoanah@hotmail.com (A.M.L.); rafakarate@hotmail.com (R.d.S.L.); wesleyhenriiique@gmail.com (W.H.d.S.); gloria@unifesp.br (M.A.d.G.); tatiana.pinotti@unifesp.br (T.P.G.); tukabc@hotmail.com (N.S.); 3Interdisciplinary Postgraduate Program in Health Sciences, Cruzeiro do Sul University, São Paulo 01506-000, Brazil; bcastino95@gmail.com; 4Graduate Program in Environmental and Experimental Pathology, Paulista University, São Paulo 04026-002, Brazil; eduardo.bondan@cruzeirodosul.edu.br

**Keywords:** mesenchymal stromal cells, kidney, omega-3 fatty acids, peroxisome-proliferator activator receptor gamma 1, allogeneic transplantation

## Abstract

Mesenchymal stromal cells (MSCs) obtained from bone marrow are a promising tool for regenerative medicine, including kidney diseases. A step forward in MSCs studies is cellular conditioning through specific minerals and vitamins. The Omega-3 fatty acids (ω3) are essential in regulating MSCs self-renewal, cell cycle, and survival. The ω3 could act as a ligand for peroxisome proliferator-activated receptor gamma (PPAR-γ). This study aimed to demonstrate that ω3 supplementation in rats could lead to the up-regulation of PPAR-γ in the MSCs. The next step was to compare the effects of these MSCs through allogeneic transplantation in rats subjected to unilateral ureteral obstruction (UUO). Independent of ω3 supplementation in the diet of the rats, the MSCs in vitro conserved differentiation capability and phenotypic characteristics. Nevertheless, MSCs obtained from the rats supplemented with ω3 stimulated an increase in the expression of PPAR-γ. After allogeneic transplantation in rats subjected to UUO, the ω3 supplementation in the rats enhanced some nephroprotective effects of the MSCs through a higher expression of antioxidant enzyme (SOD-1), anti-inflammatory marker (IL-10), and lower expression of the inflammatory marker (IL-6), and proteinuria.

## 1. Introduction

Mesenchymal stromal cells (MSCs) obtained from bone marrow were the first stem cells to be well-characterized [1] and are a promising tool for regenerative medicine. The efficacy of MSCs-based cell therapy has been demonstrated for a broad range of indications, including kidney diseases [2,3]. Progressive deterioration observed in chronic kidney disease (CKD) is the consequence of a series of inflammatory events leading to interstitial cell infiltration, glomerulosclerosis, activation of fibrogenic factors, and the formation of fibrosis [2,3]. Since the process of renal repair depends on the extent and severity of kidney injury, the development of MSCs therapy that aims to restore or replace chronically injured tissues has brought hope for the treatment of many chronic diseases. However, donor characteristics such as gender, age, diet, and health status can affect the feasibility of MSCs, resulting in positive and negative results [4,5].

A step forward in MSCs studies is cellular conditioning through diet with specific minerals and vitamins [6,7,8]. The major bioactives of omega-3 fatty acids (ω3) are eicosapentaenoic acid (EPA) and docosahexaenoic acid (DHA), which play an essential role in regulating MSCs self-renewal, cell cycle, and survival [6,7,8]. The ω3 are increasingly being recognized as essential modulators of multiple biological pathways that act in kidney disease, and the ω3 supplementation as adjunctive therapy in treating CKD is established in the literature [8] due to its anti-inflammatory and antioxidant properties.

ω3 is a ligand for peroxisome proliferator-activated receptor gamma (PPAR- γ) [9,10]. It is a steroid nuclear transcription factor that regulates the expression of multiple genes to modulate energetic metabolism, cell differentiation, and apoptosis [10,11]. PPAR- γ also plays a crucial role in kidney disease due to its anti-inflammatory, antifibrotic, and antioxidant properties [10,11].

Synthetic PPAR-γ agonists such as pioglitazone, rosiglitazone, and troglitazone have demonstrated a therapeutic effect in various kidney conditions, but their use is restricted by their associated side effects [12]. As a result, there is a growing interest in examining natural products that have the potential to activate PPAR-γ, which could be a promising approach to developing effective and safe treatments for kidney diseases. Our group previously demonstrated that the beneficial effects of isoflavones in the kidneys of obese rats have been mediated through the expression of PPAR-γ [13].

The unilateral ureteral obstruction (UUO) is an experimental model in rodents that is capable of mimicking all events of human chronic kidney disease in a relatively short time. It is characterized by tubular dilation, loss of proximal tubular mass, interstitial expansion, hydronephrosis, hypertrophy, infiltration of leukocytes, the presence of fibroblasts, and tubular renal epithelial cell death [14]. These alterations are a result of molecular processes such as fibrosis, oxidative stress, and inflammation [14]. This study aimed to demonstrate that ω3 supplementation in rats could lead to the up-regulation of PPAR-γ in the MSCs. The next step was to compare the effects of these MSCs through allogeneic transplantation in rats subjected to UUO.

## 2. Results

### 2.1. In Vitro

As shown in Figure 1, compared to the MSCs obtained from rats unsupplemented (Figure 1A), MSCs-ω3 obtained from rats supplemented with ω3 (Figure 1B) also appeared as a monolayer of large, fibroblast-like flattened adherent cells (at passage 4). After 21 days of induction, the potential differentiation into osteocytes (Figure 1C,D) and adipocytes (Figure 1D,E) was observed in both groups. Immunophenotypes were tested by flow cytometric analysis (Figure 1G). Both groups were negative for CD45 and CD34 and positive for CD73 and CD90.

Figure 2 shows the immunoblot analyses of PPAR-γ (Figure 2A) with their respective graphical quantifications (Figure 2B). The MSCs obtained from rats supplemented with ω3 showed increased expression of PPAR-γ in comparison with MSCs obtained from unsupplemented rats.

### 2.2. In Vivo

Fluorescent dye tracking was performed to assess the homing of MSCs to the left kidney (Figure 3). Since MSCs were from the bone marrow of male rats, we used the anti-Y-chromosome antibody to detect the MSCs in female rats subjected to UUO. At 48 h after MSCs injection, male cells (Y-chromosome) were observed in the renal tissue of the UUO female group, assuring MSCs homing to this tissue.

Table 1 shows kidney function parameters in each group evaluated. After seven days of right ureteral obstruction, all rats presented with left kidney hypertrophy. We observed a minimal increase in plasma creatinine, plasma urea, and urinary protein levels in the UUO compared to the sham group. However, there was a significant decrease in urinary protein levels after allogeneic transplantation of MSCs obtained from rats unsupplemented and rats supplemented with ω3 in rats subjected to UUO (versus UUO group). The MSCs-ω3 cells transplantation significantly decreased urinary protein levels when compared to the rats transplanted with MSCs (UUO-MSCs group).

The pro-inflammatory and pro-fibrotic status of the left kidney is shown in Figure 4. The UUO promoted the induction of TGF-β1 and IL-6 in the left kidney compared to the sham group. There was a significant reduction in TGF-β1 (Figure 4A,B) and IL-6 (Figure 4A,C) after allogeneic transplantation of MSCs obtained from rats unsupplemented and rats supplemented with ω3 in rats subjected to UUO when compared to rats subjected to UUO. In the case of IL-6, the administration of MSCs-ω3 obtained from rats supplemented with ω3 (UUO-MSCs-ω3 group) stimulated a decrease in their expression compared to the group transplanted only with MSCs obtained from unsupplemented rats (UUO-MSCs group).

The anti-inflammatory and antioxidant status of the left kidney is shown in Figure 5. After allogeneic transplantation of MSCs-ω3 obtained from rats supplemented with ω3 (UUO-MSCs-ω3 group) and MSCs obtained from unsupplemented rats (UUO-MSCs group), there was an increase in IL-10 (Figure 5A,B) and SOD-1 (Figure 5A,C) when compared with rats subjected to UUO.

After the allogeneic transplantation with MSCs-ω3 obtained from rats supplemented with ω3 (UUO-MSCs-ω3 group), they have increased IL-10 (Figure 5A,B) and SOD-1 (Figure 5A,C) expressions when compared to the group transplanted with MSCs obtained from unsupplemented rats (UUO-MSCs group). The HO-1 expression (Figure 5A,D) was increased in the UUO group (vs. sham) and diminished significantly in UUO+MSCs and UUO+MSCs-ω3 groups (vs. UUO).

## 3. Discussion

MSCs were obtained from rats unsupplemented, and MSCs-ω3 were obtained from rats supplemented with ω3 for 28 days. The presence of ω3 preserved the potential differentiation into osteocytes and adipocytes, with the expression of cell surface markers characteristic of MSCs.

ω3 and their metabolites can exert their biological effects via multiple mechanisms. It can be readily incorporated into membrane phospholipids, altering the physical and chemical properties of lipid rafts and cell membranes and thereby modulating the activity of membrane-associated functional proteins, such as receptors and ion channels [7,15]. More importantly, the ω3 are ligands or coactivators for critical transcriptional factors, such as PPAR-γ [9,10].

The PPAR-γ expression in the MSCs obtained from rats supplemented with ω3 was 6.5-fold higher than that in the MSCs obtained from rats unsupplemented. According to our knowledge, this is the first report to demonstrate that the up-regulation of PPAR-γ in the MSCs is influenced by diet, supporting donor characteristics’ importance in cellular transplant and also the donor’s diet.

The next step was to compare the effects of these MSCs obtained from rats unsupplemented and supplemented with ω3 through allogeneic transplantation in rats subjected to unilateral ureteral obstruction (UUO) for seven days since the initial process of inflammation and fibrosis is observed as early as three days after the procedure [16], and renal interstitial fibrosis may be reversible in the early stages [17,18,19]. First, we confirmed the successful MSCs homing to the tissue kidney in the rats subjected to UUO since allogeneic transplantation is homing dependent.

Our results showed that both MSCs decreased IL-6 expression in kidney tissue, but the MSCs obtained from rats supplemented with ω3 were more efficient (3.3-fold lower than in MSCs obtained from rats unsupplemented). It has been reported that IL-6 expression is elevated in CKD patients’ renal tissue [20]. Several clinical studies support the clinical blockade of IL-6 signaling in many renal-pro-inflammatory conditions [20,21,22].

Jin et al. [23] demonstrated that after the onset of UUO, more severe fibrosis and inflammation develop in the kidneys of rats lacking IL-10 than in control rats. Mu W et al. [24] showed in the UUO model that, in IL-10-treated rats, there was an approximately 50–60% reduction in TGF-β1 compared with control rats. Some authors have also suggested that the ability of MSCs to treat kidney diseases is partly mediated by IL-10 [25,26]. Németh et al. [26] showed that serum IL-10 levels are increased in mice treated with MSCs compared to the mice untreated. Here, the transplantation of MSCs increased IL-10 expression in kidney tissue, and the MSCs obtained from rats supplemented with ω3 were more efficient (2.4-fold higher than that in the MSCs obtained from rats unsupplemented).

As has been previously reported in the literature, lL-10 expression is involved in the regulation of the immune response by ω3 [27,28]. At the same time, the research of Kim MG et al. [29] has provided evidence that rosiglitazone (synthetic PPAR-γ agonists)-mediated nephroprotective effect in cisplatin nephrotoxicity in mice was partially mediated by upregulation of anti-inflammatory IL-10 production. Corroborating the study, we used a natural product that potentially activates PPAR-γ without the associated side effects observed in the use of synthetic PPAR-γ agonists [12]. Our findings suggest that ω3-stimulating up-regulation of PPAR-γ induced MSCs to produce more IL-10, which reduced inflammation in the kidney.

As expected, the inflammatory reaction in the obstructed kidney stimulated the expression of TGF-β1, which is widely recognized as a potent inducer of fibrosis in renal structures during UUO [14]. One factor whose functions are closely related to TGF-β1 is the antioxidant enzyme HO-1 [30]. Studies show that its regulation is complex and cell-specific; the TGF-β1 and HO-1 are robustly induced in tubular cells after UUO [30], and the upregulation of HO-1 is suggested as an adaptive response to counteract the injurious and profibrogenic properties of TGF-β1 overexpression [30]. In line with these findings, they were overexpressed in the kidney tissue of the UUO group in our study. The effect of transplantation of MSCs obtained from rats unsupplemented and supplemented with ω3 was similar; there was a lower expression of HO-1 and a lower expression of TGF-β1.

Since ROS also contributes to the fibrotic process either directly or indirectly via enhanced inflammation, studies have evaluated the effect of the antioxidant enzyme SOD-1, and this family is significantly downregulated after UUO [31]. MSCs obtained from rats unsupplemented and supplemented with ω3 exerted antioxidant properties, and the MSCs obtained from rats supplemented with ω3 were more efficient (1.4-fold higher than the MSCs obtained from rats unsupplemented).

Previous research showed that the effectiveness of mitochondrial transfer from MSCs is probably due to their capacity to enhance SOD-1 expression, which protects damaged cells from ROS [32,33,34]. In this regard, it has been shown that ω3 intake also regulates mitochondrial biogenesis through increased SOD-1 activity [35,36]. In addition, the activation of PPAR-γ also induces the expression of SOD-1 [37]. Thus, our findings suggest that the effect of MSCs was enhanced by ω3 stimulating up-regulation of PPAR-γ, but the mitochondrial transfer in this model still has to be determined.

Inflammation is a common occurrence in patients with CKD, and it is inversely linked to kidney function and positively associated with the magnitude of proteinuria excretion [38]. The decrease in inflammation is connected to a decrease in proteinuria [38]. Researchers are considering not only agents that directly reduce proteinuria, but also molecules that slow or prevent renal fibrosis. We showed a decrease in proteinuria after treatment with MSCs obtained from rats supplemented with ω3 that were more efficient than the MSCs obtained from rats unsupplemented.

For the first time, our data established that oral ω3 supplementation leads to the up-regulation of PPAR-γ in the MSCs and potentialized some effects in these cells in allogeneic transplantation in rats subjected to UUO through a higher expression of antioxidant enzyme (SOD-1), anti-inflammatory marker (IL-10), and lower expression of the inflammatory marker (IL-6). We also associated the decline in inflammation with lower proteinuria. Our findings showed that ω3 supplementation leads to the up-regulation of PPAR-γ in the MSCs, and the nephroprotective effect via allogeneic transplantation was potentiated in a model of kidney fibrosis in the early stage.

## 4. Materials and Methods

### 4.1. Animals and Treatment

The experimental protocol in vivo was performed in accordance with the Brazilian guidelines [39] for scientific animal care and use and was approved by the ethics committee (Research Ethics Committee 7926130116/2016). Male Wistar rats weighing 190–210 g at 30 days of age were housed in individual boxes with wood shavings and maintained at 22–24 °C, 10% relative humidity, and an alternating 12/12 h light/dark cycle. The rats were fed standard rat chow (Nuvilab, Colombo, Brazil) and water ad libitum. The rats were euthanized 30 days after the beginning of the experimental protocol through an intraperitoneal injection of a toxic dose of 10 mg/kg of xylazine (Agribrands do Brasil, São Paulo, Brazil) and 90 mg/kg of ketamine (Agribrands do Brasil, São Paulo, Brazil).

After acclimation for seven days, rats were randomly assigned to the control group (*n* = 5) or the ω3 group (*n* = 5). Rats were treated daily for 25 days with either vehicle (PBS; the control group) or 2 g/kg bw with ω3 containing 500 mg of DHA and 100 mg of EPA in a 1000 mg capsule (Naturalis^®^, São Paulo, Brazil) by oral gavage (ω3 group). This dose is in accordance with the World Health Organization (WHO) [40,41], which recommends regular consumption of 30% of the total calorie intake provided by lipids and 0.25 to 2 g of EPA and DHA consumption for the general population.

For 24 h, the animals were placed in metabolic cages to collect urine, and blood samples from the lateral tail vein were collected. The rats were euthanized 30 days after the beginning of the experimental protocol through an intraperitoneal injection of a toxic dose of 10 mg/kg of xylazine (Agribrands do Brasil, São Paulo, Brazil) and 90 mg/kg of ketamine (Agribrands do Brasil, São Paulo, Brazil), and both kidneys were then removed. Thus, the MSCs were obtained from rats unsupplemented, and MSCs-ω3 were obtained from rats supplemented with ω3. After 30 days, the MSCs were collected from the tibia and femur.

### 4.2. Culture and Characterization of Bone Marrow Mesenchymal Stromal Cells (MSCs)

The MSCs were collected from the tibia and femur and were isolated by the method described by Smajilagić et al. [42]. The cell culture of MSCs and MSCs-ω3 groups was grown at 37 °C in a humidified atmosphere containing 5% carbon dioxide in Dulbecco’s modified Eagle’s medium (DMEM, Sigma Chemicals, St. Louis, MO, USA), 24 mM of sodium bicarbonate, 10 mM of N’-2-hydroxyethyl piperazine- N’-2-ethane sulfonic acid, and 10.000 U/L of streptomycin/penicillin, and supplemented with 5% fetal bovine serum (FBS, Gibco, Vacaville, CA, USA).

The potential differentiation of MSCs into osteocytes was conducted. The cells were incubated for 3 weeks with 10 mM β-glycero-phosphate, 50 µg/mL ascorbic acid 2-phosphate, and 10 M dexamethasone (all from Sigma, St. Louis, MO, USA). The cells were fixed with 10% formalin for 20 min at room temperature. The presence of calcium-rich hydroxyapatite (mineralization) in the extracellular matrix was visualized with 5% of Von Kossa Silver (Sigma). Nikon fluorescence microscopy (Nikon, Tokyo, Japan) was used to show the results through photomicrographs.

The potential differentiation of MSCs into adipocytes was also performed. For 3 weeks, the cells were incubated with 5 g/mL insulin (Sigma) and 10 M dexamethasone (Sigma). Phase-contrast microscopy showed adipogenic differentiation due to the presence of highly refractive intracellular lipid vacuoles. Oil Red O (Sigma) staining was used to measure the accumulation of lipid droplets in these vacuoles. Nikon fluorescence microscopy (Nikon, Tokyo, Japan) was used to show the results through photomicrographs.

The markers of MSCs, anti-CD45, anti-CD73, and anti-CD90 (1:10, BD Biosciences, San Jose, CA, USA), were used in flow cytometric analysis, and one negative control tube with cell suspension was also used as a control. The cells were incubated with purified antibody, washed twice with PBS buffer, and incubated with anti-rabbit antibody conjugated to Alexa Fluor 488 (from Becton Dickinson Company, Franklin Lakes, NJ, USA) for 20 min. After incubation, the cells were washed with PBS buffer and then resuspended with 500 µL of PBS for FACS analysis. In duplicate, flow cytometry experiments were conducted. The Cell Quest program conducted statistical analyses of flow cytometry, and the results were expressed in percentages.

### 4.3. In Vitro Analyses of PPAR-γ

Western blotting was performed in the cells as described previously [43]. The protein concentration was verified by the method of Lowry [44]. MSCs and MSCs-ω3 cells were lysed with RIPA lysis buffer (200 μL) per plate (100 mm^2^). For 5 min at 4 °C, the lysates of cells were centrifuged at 12,000 g, and the supernatants were stored at −80 °C. 30 μg of proteins were separated by 10% polyacrylamide gel electrophoresis and transferred to polyvinylidene fluoride (PVDF) membranes using a Mini Trans-Blot Electrophoretic Transfer Cell (BioRad). The nonspecific binding sites were blocked in a TBS buffer with 5% albumin. The immunoblots were incubated overnight at 4 °C with GAPDH (1:1000, Santa Cruz Biotechnology, Dallas, TX, USA) and peroxisome-proliferator activator receptor gamma: PPAR-γ (1:500, Abcam, Cambridge, UK). After washing with TBS-T, the membranes were incubated for 1 h at 4 °C in HRP-conjugated secondary antibodies (1:30,000; Cell Signalling). Using Pierce ECL Plus Chemiluminescent substrate-detecting reagents (Thermo Fisher, Waltham, MA, USA), the immunoreactive protein bands were visualized. Images were obtained and analyzed with an Alliance 7 Chemiluminescence documentation system (UVItec, Cambridge, UK). The immunoblot band intensities were quantified using Image J software and expressed as the PPAR-γ/GAPDH ratio.

### 4.4. Homing of MSCs

Fluorescent dye tracking was performed to assess the homing of MSCs to kidney tissue. Since MSCs originated from the bone marrow of male rats, we used an anti-Y-chromosome antibody (1:50, Aviva Systems Biology, San Diego, CA, USA) to detect the cells in three female rats that received 1 × 10^6^ MSCs cells resuspended in PBS via the cava vein. Briefly, kidney tissues were fixed in acetone, frozen, and immersed in PBS for 5 min, and immunostaining was performed overnight at 4 °C. After washing with PBS, the kidney tissues were incubated for 1 h with the secondary antibody Alexa Fluor 568 (1:150, Thermo Fischer Scientific, Waltham, MA, USA). Microscopic images were obtained using Nikon fluorescence microscopy (Nikon, Tokyo, Japan). The results were shown through photomicrographs.

### 4.5. Unilateral Ureteral Obstruction and Allogeneic Transplantation

The male Wistar rats weighing between 190 and 210 g at 30 days of age were anesthetized for the UUO procedure, performed as described previously [14], and a low midline abdominal incision was made. The ureter was mobilized, isolated with minimal dissection, and ligated with two 6.0 silk sutures at the ureterovesical junction. Rats in the sham group underwent an identical surgical procedure without ureteral ligation. Soon after, for allogeneic transplantation with MSCs and MSCs-ω3, rats received 1 × 10^6^ cells that were resuspended in PBS via the cava vein. The experimental animals were separated into six groups (*n* = 5 each) for seven days: Sham, Sham+MSCs, Sham+MSCs-ω3, UUO, UUO+MSCs, and UUO+MSCs-ω3.

The animals were placed in metabolic cages for 24 h to collect urine, and blood samples were taken from the lateral tail vein. On day seven post-surgery, the animals were euthanized through an intraperitoneal injection of a toxic dose of 10 mg/kg of xylazine (Agribrands do Brasil, São Paulo, Brazil) and 90 mg/kg of ketamine (Agribrands do Brasil, São Paulo, Brazil), and both kidneys were then removed.

### 4.6. In Vivo Analyses

An AD-5000 balance (Marte Cientfica Ltd.a., São Paulo, Brazil) was used, and the measurement of body weight and kidney weight were calculated, and the results were expressed in grams.

Western blotting in the left kidney was performed as described previously [13]. The protein concentration was verified by the method of Lowry [43]. The kidney tissues were lysed with 200 μL of a RIPA lysis buffer and centrifuged at 12,000 g for 5 min at 4 °C. The tissues lysates, and supernatants were stored. The proteins were separated by 10% polyacrylamide gel electrophoresis and transferred to PVDF membranes, sing a Mini Trans-Blot Electrophoretic Transfer Cell (BioRad). In a TBS buffer, 5% albumin was used to block the nonspecific binding sites. The immunoblots were incubated overnight at 4 °C with the superoxide dismutase-1: SOD-1 (1:1000, Abcam, Cambridge, UK), interleukin-6: IL-6 (1:1000, Novus Biologicals, Centennial, CO, USA), interleukin-10: IL-10 (1:1000, Abcam, Cambridge, UK), heme oxygenase-1: HO-1 (1:1000, Abcam, Cambridge, UK), transforming growth factor beta-1: TGF-β1 (1:1000, Abcam, Cambridge, UK), and beta-actin: β-actin (1:1000, Santa Cruz Biotechnology, Dallas, TX, USA). The immunoblot band intensities were quantified using ImageJ software and expressed as a ratio of molecule/β-actin.

The urine and plasma levels of urea and creatinine were spectrophotometrically assayed according to standard procedures using commercially available diagnostic kits (Labtest Diagnóstica, Minas Gerais, Brazil). The creatinine level was established using a colorimetric method based on the Jaffé reaction [45]; the results are expressed as mg/dL. Urinary proteins were established using a colorimetric method based on pyrogallol red-molybdate [46]; the results are expressed as mg/mL.

### 4.7. Statistical Analysis

The descriptive statistical analysis of the data was performed using the Action Stat software (version 3.3.2) for Windows. Initially, the data were evaluated using the Shapiro–Wilk normality test. Then, the data with a non-normal distribution were evaluated using the Kruskal–Wallis test with subsequent application of Bonferroni criteria, while the data with a normal distribution were compared using the Tukey post hoc test. Data are reported as mean ± SEM. The significance level was set at 5% (*p* < 0.05).

## 5. Conclusions

The nephroprotective effects of allogeneic transplantation with mesenchymal stromal cells were potentiated by ω3 in rats, a natural product that has the potential to activate PPAR-γ in an animal model of kidney fibrosis that typifies forms of chronic kidney disease. The nephroprotective action of mesenchymal stromal cells could also be influenced by the donor’s lifestyle characteristics, such as diet.

## Figures and Tables

**Figure 1 pharmaceuticals-16-01484-f001:**
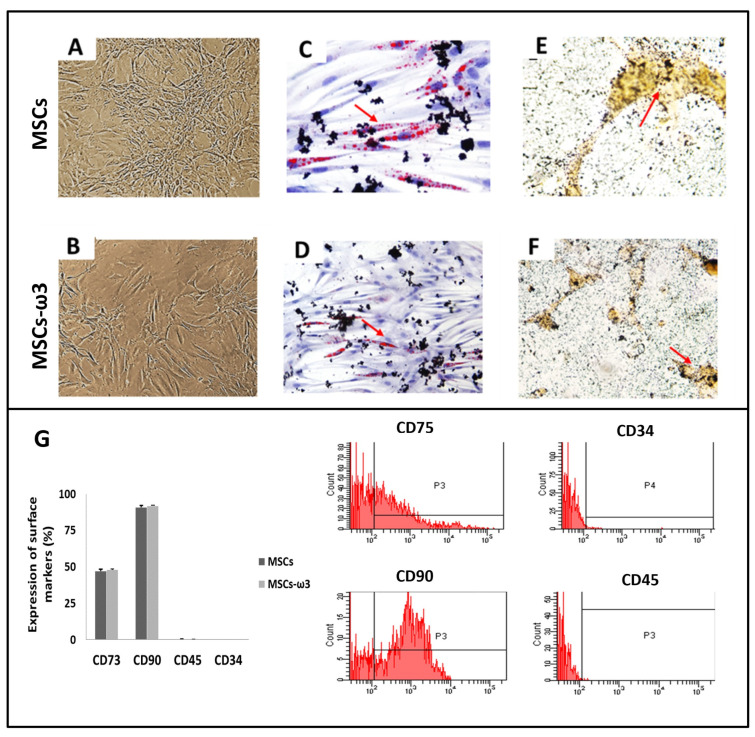
Cell differentiation assays. Representative light microscopic images (**A**,**B**) Osteogenic differentiation (**C**,**D**), stained with Alizarin Red: the arrow shows the presence of cells containing calcium. Adipogenic differentiation (**E**,**F**), stained with Oil Red O: the arrow indicates lipid droplets. Immunophenotypes of surface markers were tested by flow cytometric analysis (**G**). Data are presented as means ± standard errors. *n* = 15 for each group.

**Figure 2 pharmaceuticals-16-01484-f002:**
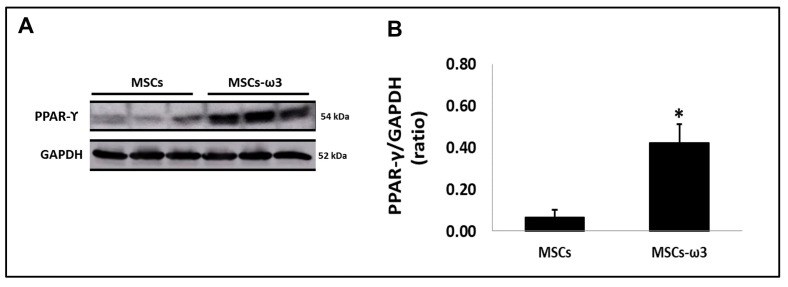
Activation of peroxisome-proliferator activator receptor gamma (PPAR-γ) in vitro. (**A**). Qualitative analysis via Western blotting. (**B**). Quantitative analyses of immunoblotting images were performed using ImageJ Version 1.52v. Data are presented as means ± standard errors. *n* = 15 for each group. (*) Indicates significant differences compared with the MSCs group at *p* < 0.05.

**Figure 3 pharmaceuticals-16-01484-f003:**
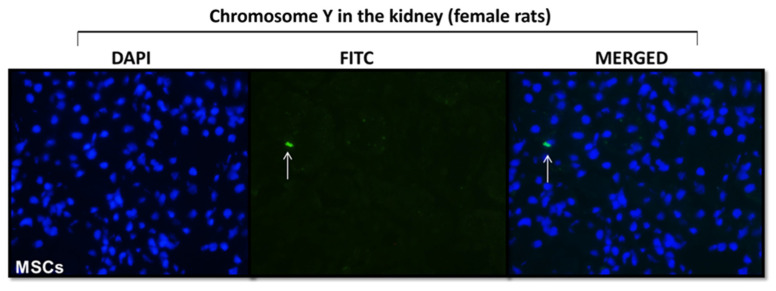
MSCs homing. Photomicrography of renal tissue staining with chromosome Y antibody. The white arrow indicates the presence of male MSCs in the kidney of females submitted to unilateral ureteral obstruction (*n* = 2).

**Figure 4 pharmaceuticals-16-01484-f004:**
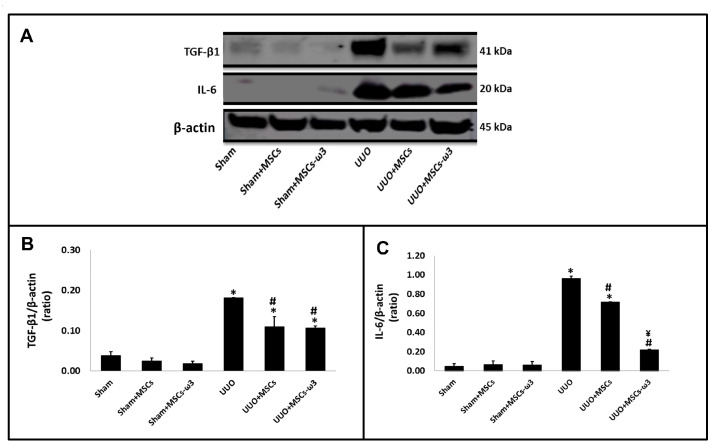
Pro-inflammatory and pro-fibrotic status of the left kidney. Western blot images (**A**) for IL-6 and TGF-β1. Quantitative analyses of immunoblotting images were performed using ImageJ for TGF-β1 (**B**) and IL-6 (**C**). Data are presented as means ± standard errors. *n* = 5 for each group. The significance level for a null hypothesis was set at 5% (*p* < 0.05). (*) All groups compared to the sham group, (#) UUO+MSCs-ω3 and UUO+MSCs groups compared to the UUO group, and (¥) UUO+MSCs-ω3 group compared to the UUO+MSCs group.

**Figure 5 pharmaceuticals-16-01484-f005:**
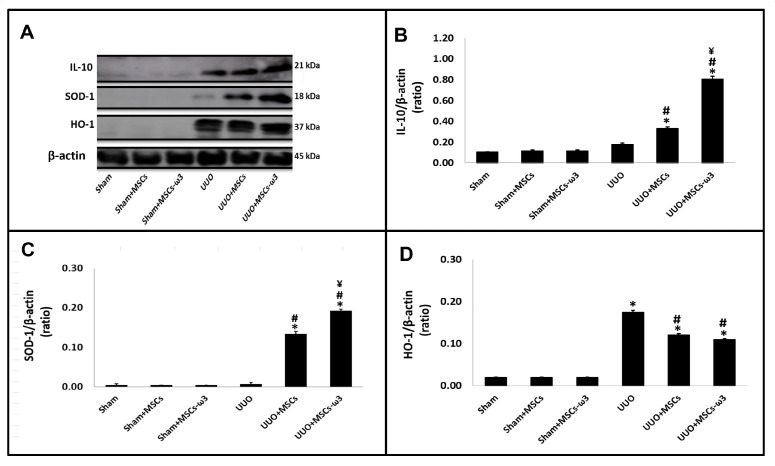
Anti-inflammatory and antioxidant status of the left kidney. Western blot images (**A**) IL-10, SOD-1, and HO-1. Quantitative analyses of immunoblotting images were performed using ImageJ for IL-10 (**B**), SOD-1 (**C**), and HO-1 (**D**). Data are presented as means ± standard errors. *n* = 5 for each group. The significance level for a null hypothesis was set at 5% (*p* < 0.05). (*) All groups compared to the sham group, (#) UUO+MSCs-ω3 and UUO+MSCs groups compared to the UUO group, and (¥) UUO+MSCs-ω3 group compared to the UUO+MSCs group.

**Table 1 pharmaceuticals-16-01484-t001:** Evaluation of renal function. Plasma creatinine, plasma urea, and urinary protein/creatinine urinary. Data are presented as means ± standard errors. *n* = 5 for each group. (*) All groups compared to the respective Sham group, (**) UUO+MSCs-ω3 and UUO+MSCs groups compared to the UUO group, (¥) UUO+MSCs-ω3 group compared to the UUO+MSCs group, and (#) left kidney compared to the right kidney.

Groups	Plasma Creatinine (mg/mL)	Plasma Urea (mg/mL)	Urinary Protein (mg/mL)	Kidney (Mean Right/g)	Kidney (Mean Left/g)
Sham	0.54 ± 0.02	36.67 ± 3.87	1.29 ± 0.07	1.207 ± 0.06	1.192 ± 0.04
Sham-MSCs	0.64 ± 0.03	39.83 ± 1.72	0.98 ± 0.10	1.249 ± 0.0	1.216 ± 0.04
Sham-MSCs-ω3	0.50 ± 0.01	40.40 ± 0.74	1.02 ± 0.08	1.110 ± 0.01	1.099 ± 0.02
UUO	0.61 ± 0.03 *	45.33 ± 1.50 *	1.89 ± 0.19 *	1.516 ± 0.08	1.934 ± 0.11 #
UUO-MSCs	0.62 ± 0.03 *	45.50 ± 1.76 *	1.36 ± 0.09 **	1.307 ± 0.04	1.853 ± 0.07 #
UUO-MSCs-ω3	0.61 ± 0.02 *	46.00 ± 1.79 *	0.92 ± 0.14 ** ¥	1.332 ± 0.03	2.047 ± 0.16 #

## Data Availability

Derived data supporting the findings of this study are available from the corresponding author [FTB] on request.

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
