# Peer review of "The Nephroprotective Effects of the Allogeneic Transplantation with Mesenchymal Stromal Cells Were Potentiated by ω3 Stimulating Up-Regulation of the PPAR-γ"

_pharmaceuticals, 2023, doi:10.3390/ph16101484_

Round 1

Reviewer 1 Report

The manuscript demonstrated that the nephroprotective role of of the MSCs is enhanced by ω3 supplementation through increased expression of antioxidants and anti-inflammatory markers and decreased expression of the inflammatory markers.

My cooments

-Figure 1 for flow cytometry, curves should be presented

-In figure 4 and 5, the significance of UUO+MSCs-ω3 and UUO+MSCs groups compared to the UUO group ** is confusing with * for comparison with the sham group, please choose another symbol

-Materials and methods section needs extensive improvement. Subtitles should be added and detailed methodology should be provided

 Minor editing of English language required

Reviewer 2 Report

Current study aimed to demonstrate that omega-3 fatty acids (ω3) supplementation in rats could lead to the up-regulation of PPAR-γ in the Mesenchymal Stromal Cells (MSCs) and the effects of these MSCs were compared through allogeneic transplantation in rats subjected to unilateral ureteral obstruction (UUO). I like to give the following comments.

1.      Unilateral ureteral obstruction (UUO) is related to what kind of renal disorders in clinic? Please describe in clear.

2.      In line 20, up-regulation of PPAR-y in the MSCs that needs to correct as gamma.

3.      In Figure 2, Western blots shown the indicator as PPAR-γ 1 and legends mentioned PPAR-γ only. Please revise it.

4.      Sample size must indicate in the legends of each figure.

5.      In Table 1, the weight of kidney needs the clear unit, such as wet weight (g) or others.

6.      In Figure 5, data from Western blots failed to reach the same as column. Please check it carefully.

7.      The higher expression of antioxidant enzyme (SOD-1) and anti-inflammatory marker (IL-10) belonged to the main mechanism(s) in UUO. It needs more reference(s) to support.

8.      The 2 g/kg of omega-3 fatty acids (ω3) supplementation treated for 25 days that needs reference(s) to support.

9.      Novelty was not mentioned in the conclusion.

It seems better to check through the professional editing.

Round 2

Reviewer 1 Report

The authors appropriately revised the manuscript and it can be published 

The authors appropriately revised the manuscript and it can be published